# Comparing Heart Rate and Heart Rate Reserve for Accurate Energy Expenditure Prediction Against Direct Measurement

**DOI:** 10.3390/ijerph22101539

**Published:** 2025-10-08

**Authors:** Yongsuk Seo, Yunbin Lee, Dae Taek Lee

**Affiliations:** Exercise Physiology Laboratory, Kookmin University, Seoul 02707, Republic of Korea; yseokss@kookmin.ac.kr

**Keywords:** energy metabolism, cardiovascular response, physiological monitoring, exercise intensity

## Abstract

This study developed and validated simplified, individualized heart rate (HR)-based regression models to predict energy expenditure (EE) during treadmill exercise without direct VO2 calibration, addressing the need for more practical and accurate methods that overcome limitations of existing predictions and facilitate precise EE estimation outside specialized laboratory conditions. Energy expenditure was measured by assessing oxygen uptake (VO_2_) using a portable gas analyzer and predicted across three treadmill protocols: Bruce, Modified Bruce, and Progressive Speed. These protocols were selected to capture a wide range of exercise intensities and improve the accuracy of heart rate-based EE predictions. The six models combined heart rate, heart rate reserve (HRres), and demographic variables (sex, age, BMI, resting HR) using the Enter method of multiple regression, where all variables were included simultaneously to enhance the real-world applicability of the energy expenditure predictions. All models showed high accuracy with *R*^2^ values between 0.80 and 0.89, and there were no significant differences between measured and predicted energy expenditure (*p* ≥ 0.05). HRres-based models outperformed others at submaximal intensities and remained consistent across sex, weight, BMI, and resting HR variations. By incorporating individual resting and maximal HR values, HRres models offer a personalized, physiologically relevant estimation method. These results support integrating HRres-based EE prediction into wearable devices to improve accessible and precise monitoring of physiological energy metabolism.

## 1. Introduction

Regular physical activity (PA) is essential for promoting health, maintaining physical fitness, and reducing the risk of chronic diseases such as cardiovascular disease, type 2 diabetes, and obesity [1]. As the global population ages, the role of PA in supporting healthy aging, preserving mobility, and improving quality of life becomes increasingly significant. However, individuals exhibit substantial variability in physiological responses to exercise, influenced by factors such as age, sex, baseline fitness, and body composition [2,3,4,5]. To address this variability, contemporary exercise guidelines advocate personalized exercise prescriptions that align with individual physiological profiles [1]. Although estimating energy expenditure (EE) plays an important role in exercise monitoring and customizing exercise intensity, it is not the sole basis for exercise prescription. Exercise prescriptions are typically guided by multiple factors including fitness goals, health status, and evidence-based guidelines focusing on intensity, duration, and frequency rather than EE alone. This study emphasizes improving EE estimation as one component to enhance personalized exercise monitoring and guidance. While direct calorimetry using a metabolic chamber is considered the gold standard for assessing EE, it is not feasible in most research or applied settings [6]. Consequently, indirect calorimetry based on oxygen uptake (VO_2_) is widely used as the practical reference method for EE assessment, as it strongly correlates with direct calorimetry [7,8]. In contrast, heart rate (HR) is widely accessible through wearable technologies and exhibits a strong linear relationship with VO_2_, making it a practical surrogate for estimating EE [9].

Various models have been developed to predict energy expenditure EE based on HR. Many of these models utilize simplified approaches, such as the net HR and HR index models, which estimate EE expressed in metabolic equivalents to standardize exercise intensity. Despite their usability, these models often show inconsistent accuracy across different populations and exercise intensities [10,11,12,13]. Moreover, while earlier studies indicated that generalized algorithms used in commercial fitness trackers often overlooked interindividual differences, resulting in notable inaccuracies, particularly during variable or nonlinear movement patterns, recent advances in sensor technology and machine learning have significantly improved algorithm accuracy. However, despite these improvements, challenges persist in accurately capturing diverse movement patterns and individual-specific factors across different populations, warranting ongoing development and validation. [14,15,16]. To address these limitations, efforts have explored incorporating HRres and HRrest into predictive models to enhance individual accuracy without requiring VO_2_ measurement [17]. The comparative validity and practical utility of HR- and HRres-based models remain underexplored, particularly when compared to direct measurements of EE across varying exercise intensities. Current HR-based prediction models often lack sufficient individualization and exhibit variable accuracy, limiting their practical application. The main objective of this study is to rigorously compare the accuracy and practical utility of heart rate (HR) and heart rate reserve (HRres)-based models for predicting energy expenditure against direct measurements across a wide range of exercise intensities. This comparison aims to address limitations related to individualization and variable accuracy in existing models, thereby contributing to the development of more precise and applicable methodologies for exercise prescription.

It is hypothesized that there is a significant difference in the accuracy of predicted energy expenditure between the HR- and HRres-based models when compared to measured energy expenditure across exercise intensities

## 2. Materials and Methods

### 2.1. Participants

Prior to partaking in the study, volunteers were screened through a health history questionnaire and the Physical Activity Readiness Questionnaire.

Participants were eligible if they were aged between 18 and 45 years and met the minimum physical activity guideline of at least 150 min of moderate-intensity exercise per week. To ensure safe participation, all potential volunteers completed the Physical Activity Readiness Questionnaire (PAR-Q) as a pre-exercise screening tool. Individuals were excluded if they smoked, had a history of cardiovascular, metabolic, or pulmonary diseases, or any condition contraindicated for exercise. Fitness level was thus defined as meeting or exceeding the 150 min weekly physical activity threshold, without requiring a higher performance level or specific fitness testing beyond the International Physical Activity Questionnaire (IPAQ) screening.

After fully understanding the study procedures, volunteers provided written and verbal informed consent to participate. Before the testing session, volunteers were instructed to abstain from alcohol, caffeine, and strenuous exercise for at least 24 h to reduce changes in heart rate and metabolism.

A total of 26 healthy adults (19 men and 7 women) participated across three distinct treadmill exercise protocols. Experiment 1 included 12 men and 4 women; Experiment 2 included 4 men; and Experiment 3 included 3 men and 3 women (Table 1). Each participant completed one of three experiments which included three treadmill exercise protocols to evaluate a comprehensive range of physiological responses.

The study was approved by the Institutional Review Board at Kookmin University (IRB No. KMU-202409-HR-431; approval date: 21 November 2024) and conducted in accordance with the Declaration of Helsinki.

### 2.2. Procedures

Before participation, all volunteers were familiarized with the study protocol. A detailed explanation of the study’s purpose and procedures was provided to ensure full understanding. Subsequently, volunteers provided written and verbal informed consent to participate.

For all experiments, the subjects wore cotton shorts and t-shirts, instrumented with HR monitor, and underwent a 10 min resting stabilization under thermoneutral conditions (20 °C and 50% relative humidity (RH)).

Experiment 1. Bruce (treadmill) protocol

Subjects (n = 16) performed a Bruce treadmill test until exhaustion (corroborated by an age-predicted maximal HR or a plateau in VO_2_) in a thermoneutral environment. The Bruce treadmill test is a maximal graded test increasing both speed and incline to assess cardiorespiratory endurance [18].

Experiment 2. Modified Bruce (treadmill) Protocol

Subjects (n = 4) performed a modified Bruce treadmill test until exhaustion (corroborated by an age-predicted maximal HR or a plateau in VO_2_) in a thermoneutral environment. The modified Bruce treadmill test follows the same structure as the Bruce treadmill test but begins with lower speeds and inclines, making it more suitable for individuals with lower fitness levels [18].

Experiment 3. Progressive Speed Treadmill Exercise

Subjects (n = 6) performed a progressive speed treadmill exercise, the progressive speed protocol increases treadmill speed incrementally (5, 7, 9, 10, 11 km/h), each for 2 min, at a constant 0% incline to isolate speed-based physiological responses.

It is of importance to note that these three experiments were originally conducted with different primary objectives, distinct from those explored in the current analysis. However, pooling these datasets enabled a comprehensive investigation of the hypotheses addressed in this paper. Utilizing multiple exercise protocols allowed for the evaluation of physiological responses across a broader and more varied range of exercise intensities, thereby enhancing the external validity and generalizability of relationships.

The measurement techniques for all variables and environmental conditions were the same for all three experimental trials. HR was continuously monitored throughout testing using a chest-strap device (Polar M2, Polar Electro Oy, Kempele, Finland), and VO_2_ was measured via a portable gas analyzer (K5, Cosmed, Rome, Italy) using a face mask with integrated flowmeter and O_2_ sensor. The age, height and weight were measured using a stadiometer (DS-102, JENIX, Seoul, Republic of Korea) and a digital weighing scale (DB-150, CAS, Yangju-si, Gyeonggi-do, Republic of Korea), respectively. Body mass index (BMI) was subsequently calculated from these values. During each experiment, HR and VO_2_ were measured continuously and averaged over 60 s intervals. This process resulted in a nested dataset with multiple observations for each individual over time.

### 2.3. Data Analysis

Statistical analyses were performed using Statistical Package for the Social Sciences 22.0 (IBM-SPSS, Somers, NY, USA). To examine the relationship between HR and HRres with measured EE, Pearson’s correlation coefficient was calculated. The data were combined to enable a more diverse and externally valid analysis of the variables of interest across different exercise settings. The required sample size to detect a statistically significant Pearson correlation coefficient was calculated using G*Power 3.1 [19]. Based on an anticipated large effect size (*R*^2^ = 0.800, equivalent to r = 0.894), a desired statistical power of 0.80, and a two-tailed significance level of α = 0.05, the G*Power exact test for Pearson’s r indicated that a minimum of 12 volunteers was sufficient. The sample size calculation was originally powered to detect meaningful physiological differences in Experiment 1 with 12 volunteers. Converting this correlation effect size to the regression context using the following equation [20]:$$f^{2}=\frac{R^{2}}{1-R^{2}}$$
where *f*^2^ is effect size and *R*^2^ is coefficient of determination

This yields an effect size of *f*^2^ = 4.0. With α = 0.05, power of 0.80, and 6 predictors, the minimal total sample size required was calculated as 7 volunteers, indicating sufficient power to detect meaningful associations. To account for potential attrition, biological variability, and to improve the generalizability of findings, we aimed to recruit a larger sample. To further assess the consistency and predictive accuracy of the models identified as most accurate within their respective categories, comparisons with metabolic energy expenditure were conducted across incremental exercise intensity intervals of approximately 10% using different exercise protocols. This approach allowed evaluation of model performance under diverse physiological conditions despite variability introduced by multiple protocols. Multiple linear regression using the Enter method was conducted to develop prediction equations for EE, with six predictors: age, height, weight, HRrest, HRmax, and HRres.

To evaluate the accuracy of the regression-derived EE predictions, a paired sample *t*-test was used to compare predicted EE values with measurements of EE obtained from a portable wireless metabolic gas analyzer. To further assess the consistency and predictive accuracy of the models across varying exercise intensities, identified as the most accurate within their respective categories, were compared with measurements of EE across incremental intensity intervals of approximately 10%. The level of statistical significance was set at α ≤ 0.05 and all data are presented as mean ± standard deviation (SD).

## 3. Results

### 3.1. Correlation Between HR, HRres, and Energy Expenditure

Figure 1 illustrates the relationships between HR, HRres, and measurements of EE. Both HR and HRres exhibited strong positive correlations with measurements of EE, with HRres showing a slightly higher correlation coefficient (*R*^2^ = 0.800) compared to HR (*R*^2^ = 0.782) (Figure 1).

### 3.2. Development of Regression Models for EE Prediction

Multiple linear regression analyses were performed to develop predicted energy expenditure models using HR or HRres as the primary independent variable, along with covariates such as sex, weight or BMI, and resting HR. As summarized in Table 1, all six models demonstrated high predictive power. HRres and HR were consistently the strongest predictors of EE (β = 0.893–0.909, all *p* ≤ 0.001). Although sex showed a negative association and weight or BMI had smaller positive effects, resting HR contributed significantly only to the HR-based models (Models 2, 4, and 6). Simplified models using only HR or HRres with resting HR (Models 5 and 6) still maintained high accuracy, indicating the robustness of HR–based prediction equations.

### 3.3. Model Accuracy Compared to Measured EE

Predicted energy expenditure values from Models 1 through 6 were compared to measured EE using paired sample *t*-tests. As shown in Table 2, none of the models differed significantly from the measured values (all *p* ≥ 0.05), indicating that the regression models reliably predicted energy expenditure (Table 3).

### 3.4. Comparisons Between HR- and HRres-Based Models

Further comparisons were made between models using HRres and HR while holding other variables constant. Results indicated no statistically significant differences between HRres- and HR-based models (e.g., Model 1 vs. Model 2; Model 3 vs. Model 4; Model 5 vs. Model 6), suggesting that both HR metrics can yield similarly accurate predictions when applied within a consistent model framework (Table 4).

### 3.5. Consistency of Predicted Energy Expenditure Within HRres- or HR-Based Models

To examine the consistency of predicted energy expenditure within each HR metric category, pairwise comparisons were conducted among models using HRres and HR. No significant differences were found among HRres-based models (Models 1, 3, and 5) or HR-based models (Models 2, 4, and 6), further supporting the reliability of the predictive equations regardless of the specific covariates included (Table 5).

### 3.6. Prediction Accuracy Across Exercise Intensities (10~100%)

Repeated measures ANOVA revealed statistically significant main effects of EE condition (Measured EE, HRres-based model, and HR-based model) across most exercise intensities, indicating notable discrepancies in energy expenditure estimation. At 10% intensity, the difference was significant (F(2, 36) = 286.014, *p* ≤ 0.001, η^2^_p_ = 0.941), and similarly at 20% (F(2, 50) = 449.634, *p* < 0.001, η^2^_p_ = 0.947) and 30% (F(2, 50) = 451.228, *p* ≤ 0.001, η^2^_p_ = 0.948). Significant effects of EE condition persisted at 40% (F(2, 50) = 387.999, *p* ≤ 0.001, η^2^_p_ = 0.939), 50% (F(2, 50) = 223.348, *p* ≤ 0.001, η^2^_p_ = 0.899), and 60% intensity (F(2, 50) = 165.915, *p* ≤ 0.001, η^2^_p_ = 0.869). At 70 and 80% intensities, differences remained significant (F(2, 48) = 71.207, *p* ≤ 0.001, η^2^_p_ = 0.748; F(2, 50) = 27.794, *p* ≤.001, η^2^_p_ = 0.526, respectively). Even at 90% intensity, a significant main effect of EE condition was observed (F(2, 46) = 10.200, *p* ≤ 0.001, η^2^_p_ = 0.307). However, at 100% intensity, no significant difference was found among the EE conditions (F(2, 22) = 0.287, *p* = 0.754, η^2^_p_ = 0.025), suggesting convergence in prediction accuracy at maximal effort. Comparison of Model 1 and Model 2 with the measured EE revealed distinct patterns across intensity levels. Model 1 showed statistically significant differences from Measured EE at 30% (*p* = 0.005), 40% (*p* < 0.001), 50% (*p* = 0.003), and 60% (*p* = 0.015), with the greatest divergence occurring at 40%. In contrast, Model 2 exhibited highly significant underestimations (*p* < 0.001) at all intensities except 100% (*p* = 0.662). These findings demonstrate that the HRres-based model consistently provided more accurate and reliable predicted energy expenditure than the HR-based model across submaximal workloads (Figure 2).

## 4. Discussion

This study aimed to evaluate the predictive accuracy of HR- and HRres-based models for predicted energy expenditure across varying exercise intensities. Both approaches demonstrated strong correlations with measured EE, with HRres models showing a modest but consistent advantage. Regression models incorporating HR or HRres alongside demographic and physiological variables yielded high predictive accuracy, supporting the validity of HR-derived metrics for predicted energy expenditure.

### 4.1. Submaximal Intensities (10~70%)

At low to moderate exercise intensities, HRres-based models consistently outperformed HR-based models. This finding aligns with prior research emphasizing individualized calibration methods [10,21]. HRres, which incorporates resting and maximal HR, provides a normalized representation of cardiovascular effort, improving model sensitivity to interindividual variability. By contrast, raw HR shows limitations at submaximal workloads due to the nonlinear relationship between HR and metabolic demand. Cardiac output and oxygen uptake increase disproportionately at these intensities, often leading to underestimation of EE [17]. Incorporating variables such as age, VO_2_max, and body composition into models has been shown to improve prediction, reinforcing the benefit of HRres as a normalization strategy.

Recent advances in modeling have highlighted the significant value of integrating multimodal data inputs to improve EE estimation. For instance, neural network models that combine heart rate (HR) and respiratory data have demonstrated a reduction in EE estimation errors by 22–60% compared to traditional models, illustrating a marked improvement in accuracy [22]. Similarly, approaches that incorporate both HR and accelerometer data have been shown to enhance prediction accuracy across various intensities and activity types [23].

### 4.2. Maximal Intensities (80~100%)

As exercise intensity approaches maximal levels, the gap between HR and HRres model performance narrows. Both methods closely approximated Measured EE at intensities between 80 and 100% VO_2_max, reflecting the increasingly linear HR–VO_2_ relationship under maximal physiological strain [10,11,24]. However, minor overestimations have been reported with HR models at 60~80% intensity levels, suggesting residual error even under near-linear conditions [17,25,26]. These findings suggest the need for further refinement of models to reduce prediction error across the full range of exercise intensities.

### 4.3. Model Structure and Consistency

Consistent performance across all HRres (Models 1, 3, 5) and HR (Models 2, 4, 6) variants confirms that incorporating variables such as BMI and sex improves model flexibility without compromising accuracy. This adaptability is especially important for wearable and field-based applications where access to physiological testing is limited. While BMI accounts for general interindividual variation in metabolic rate and enhances prediction accuracy in both accelerometer and HR-hybrid models [27,28,29], substantial evidence demonstrates that fat-free mass (FFM) is a stronger determinant of resting metabolic rate and energy expenditure than fat mass alone, and is closely linked to differences in energy intake and expenditure across populations [30].

Similarly, sex-specific calibration has been shown to reduce predicted energy expenditure error by up to 18% during aerobic activities, reflecting differences in fat-free mass and HR–VO_2_ kinetics [31]. Furthermore, participant fitness level and physical activity status are critical for model generalizability, as fitness level and demographic factors can influence energy expenditure, metabolic health, and response to physical activity interventions [32,33]. These simplified yet accurate models are well-suited for integration into mobile health platforms and consumer wearables. Devices like Fitbit and Apple Watch already achieve reasonable EE estimates using only HR and basic demographics [14].

### 4.4. Strengths

HRres-based models provide a practical and accessible approach for estimating EE in a variety of real-world settings, including clinical environments, athletic training programs, and general fitness routines. Because they do not require direct calorimetry or gas exchange analysis, these models are especially suited for continuous, personalized health monitoring through wearable technologies. In practice, resting heart rate, easily measured in a seated position for one minute before activity, can serve as a convenient indicator of individual fitness level [34,35,36], potentially substituting for more complex measures such as VO_2_max. Additionally, incorporating age-predicted maximum heart rate, a standard variable in energy expenditure prediction [17,34], can enhance model accuracy by accounting for age-related differences in resting metabolic rate [35]. These practical features support the broader application of HRres-based algorithms in personalized exercise planning and health intervention design.

### 4.5. Limitation

Several limitations should be noted. First, the study sample consisted solely of healthy adults, which may limit the generalizability of the findings to populations with cardiovascular, metabolic, or age-related conditions. Second, this study’s design was limited because the three experiments were originally conducted for different primary purposes than those addressed in the present analysis. While combining these datasets enhanced the external validity of the findings, future research should use purpose-built protocols specifically designed to test these preliminary hypotheses. Additionally, the regression models were both developed and validated within the same dataset, raising concerns about potential overfitting and limiting the ability to fully assess the models’ generalizability. Although this study provides initial evidence for the model’s predictive performance within the examined cohort, further validation is needed before it can be confidently applied to other populations. Future research would need to test the model’s generalizability using diverse, independent datasets, which is expected to strengthen its reliability and enhance its clinical or scientific utility. Third, maximal HR used to compute HRres was estimated with age-predicted formulas rather than directly measured values, potentially introducing individual-level error. Future studies should incorporate age-, sex-, and condition-specific modeling approaches and consider nonlinear or machine learning algorithms to improve predictive accuracy. Methodologically, integrating HR with additional physiological signals, such as respiration or accelerometry (sensor fusion), may further enhance model precision across varying exercise intensities. Finally, activity-specific calibration remains essential, as EE differs by exercise modality; for example, treadmill walking typically results in higher EE than cycling or rowing.

## 5. Conclusions

While both HR and HRres are effective predictors of EE, HRres-based models demonstrate superior predictive accuracy during submaximal exercise. This is because HRres methods account for individual differences in resting and maximal heart rates, leading to more personalized and physiologically meaningful EE estimates. For practical application, integrating HRres-based estimation into wearable devices and digital health platforms could make metabolic monitoring more accessible and accurate for a wider range of users. By encouraging users to input or measure their true resting and maximal heart rates, these technologies can deliver more tailored feedback, support individualized coaching, and inform activity recommendations.

## Figures and Tables

**Figure 1 ijerph-22-01539-f001:**
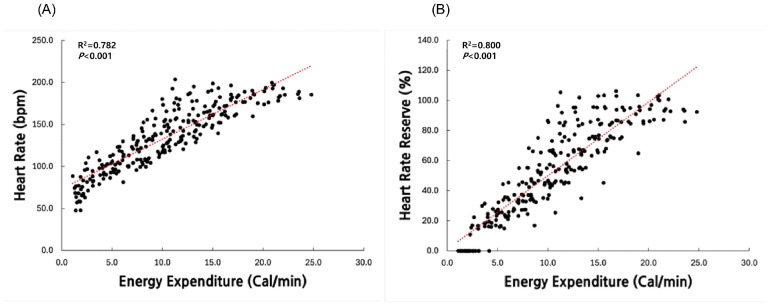
Correlation between (**A**) heart rate (HR) and (**B**) heart rate reserve (HRres) with measured energy expenditure.

**Figure 2 ijerph-22-01539-f002:**
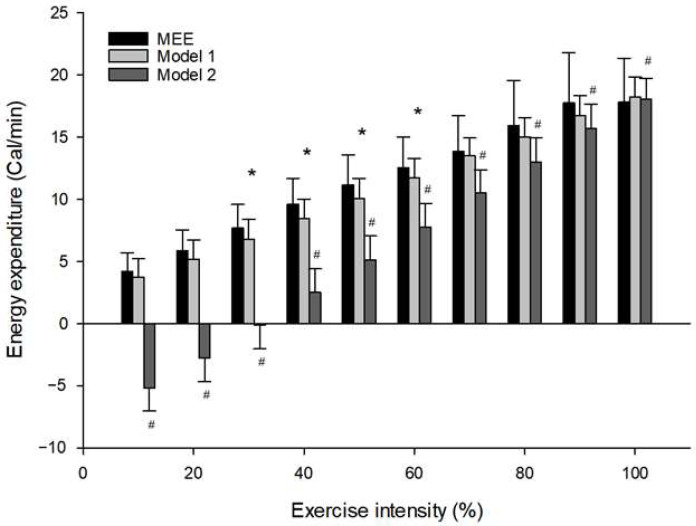
Incremental Exercise Intensity: Comparison of Measured and Predicted Energy Expenditure. * Significantly different from Measured EE. ^#^ Significantly different from MEE. MEE—measured energy expenditure.

**Table 1 ijerph-22-01539-t001:** Participant characteristics by experiment and sex.

Experiments	Group	n	Age (yrs)	Weight (kg)	Height (cm)	BMI (kg/m^2^)
Experiment 1	Total	16	28.0 ± 5.4	73.1 ± 10.0	171.4 ± 7.2	24.8 ± 2.5
	Men	12	27.8 ± 3.3	77.0 ± 10.3	174.1 ± 5.3	25.4 ± 2.9
	Women	4	28.5 ± 3.3	61.4 ± 10.3	163.3 ± 5.3	23.0 ± 2.9
Experiment 2	Total	4	25.8 ± 4.3	71.2 ± 7.1	175.3 ± 2.2	23.1 ± 1.9
	Men	4	25.8 ± 4.3	71.2 ± 7.1	175.3 ± 2.2	23.1 ± 1.9
	Women	-	N/A	N/A	N/A	N/A
Experiment 3	Total	6	26.2 ± 1.7	66.9 ± 8.8	167.6 ± 3.6	23.7 ± 2.5
	Men	3	27.3 ± 0.6	75.1 ± 2.6	170.2 ± 3.0	25.9 ± 1.6
	Women	3	25.0 ± 2.0	58.7 ± 4.9	165.0 ± 3.1	21.5 ± 1.0
Total	Total	26	27.2 ± 4.7	71.4 ± 9.7	171.1 ± 6.4	24.3 ± 2.5
Men	Men	19	27.3 ± 5.2	75.5 ± 6.8	173.7 ± 5.0	25.0 ± 2.2
Women	Women	7	27.0 ± 3.0	60.3 ± 7.4	164.0 ± 3.9	22.3 ± 2.1

Values are Mean ± Standard deviation.

**Table 2 ijerph-22-01539-t002:** Regression Models for Energy Expenditure (EE) Prediction.

Model	Primary Predictor	Other Variables	EE Prediction Equation	β	*p*
1	HRres	Sex, Weight, HRrest	EE = 0.051 + (0.164 × HRres) − (2.271 × S) + (0.075 × W) − 0.009 × HRrest	0.899	≤0.001
2	HR	Sex, Weight, HRrest	EE = −5.944 + (0.136 × HR) − (2.363 × S) + (0.077 × W) − (0.063 × HRrest)	0.909	≤0.001
3	HRres	Sex, BMI, HRrest	EE = 1.923 + (0.164 × HRres) − (2.964 × S) + (0.167 × BMI) − (0.005 × HRrest)	0.899	≤0.001
4	HR	Sex, BMI, HRrest	EE = −3.140 + (0.136 × HR) − (3.167 × S) + (0.141 × BMI) − (0.058 × HRrest)	0.909	≤0.001
5	HRres	HRrest	EE = 2.709 + (0.163 × HRres) − (0.011 × HRrest)	0.893	≤0.001
6	HR	HRrest	EE = −3.114 + (0.135 × HR) − (0.065 × Hrrest)	0.900	≤0.001

EE, energy expenditure in kcal/min; HR, heart rate; HRres, heart rate reserve; HRrest, resting heart rate; W, weight; BMI, body mass index; S, sex coded as M = 1, F = 2.

**Table 3 ijerph-22-01539-t003:** Comparison of Measured EE and model predictions (Cal/min).

Group	Mean Difference (Cal/min)	*t*	*p*
MEE—Model 1	0.1 ± 0.6	0.329	0.745
MEE—Model 2	0.1 ± 0.6	0.406	0.688
MEE—Model 3	0.1 ± 0.6	0.353	0.727
MEE—Model 4	0.0 ± 0.6	0.101	0.920
MEE—Model 5	0.1 ± 0.6	0.221	0.827
MEE—Model 6	0.1 ± 0.6	0.125	0.902

Values are mean ± SD. MEE, measured energy expenditure.

**Table 4 ijerph-22-01539-t004:** Comparison of Energy Expenditure Predictions: HRres vs. HR Models.

Group	Mean Difference (Cal/min)			*t*	*p*
Model 1—Model 2	0.0	±	0.6	0.205	0.839
Model 3—Model 4	−0.1	±	0.6	−0.928	0.362
Model 5—Model 6	0.0	±	0.4	−0.578	0.569

Values are mean ± SD.

**Table 5 ijerph-22-01539-t005:** Comparison of predicted energy expenditure: HRres vs. HR Models.

Group		Mean Difference ± SD (Kcal/min)	*t*	*p*
HRres Models	Model 1—Model 3	0.0 ± 0.6	0.214	0.832
	Model 1—Model 5	0.0 ± 0.5	−0.051	0.960
	Model 3—Model 5	0.0 ± 0.5	−0.011	0.991
HR Models	Model 2—Model 4	−0.1 ± 0.6	−1.155	0.259
	Model 2—Model 6	0.0 ± 0.5	0.136	0.893
	Model 4—Model 6	0.0 ± 0.5	−0.095	0.925

Values are mean ± SD. HR, heart rate; HRres, heart rate reserve.

## Data Availability

The data presented in this study are available on request from the corresponding author. The data are not publicly available due to the protection of personal information.

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
