# Peer review of "Comparing Heart Rate and Heart Rate Reserve for Accurate Energy Expenditure Prediction Against Direct Measurement"

_ijerph, 2025, doi:10.3390/ijerph22101539_

Round 1
Reviewer 1 Report
Comments and Suggestions for Authors
Abstract: It is not clear in the abstract what the criterion measure was to validate the regression models. Some additional results including participant demographics would also be helpful in the abstract for context.
Introduction: is clear and concise, however I would challenge the authors that VO2 is the gold standard for measuring EE. Technically, a metabolic chamber is the gold standard for assessing EE, however this is not feasible for the majority of researchers, therefore the indirect method of VO2 is utilised, and correlated well with the direct method. I would advise a slight reword and clarification of the direct v indirect and gold standard of EE assessment.
Methods: could you state more explicitly the inclusion criteria for participating in the study i.e. age range, fitness level (i.e. did they just need to meet 150 min PA per week or did you require a higher level of fitness). Would also be helpful to reference the pre-exercise screening tool utilised.
Why was there varying numbers of participants who completed each protocol? A much stronger methodology would have been a cross over design. Some justification as to how participants were allocated is warranted, and with the sample size calculation, was 12 the number of participants required per group completing each protocol? It is not clear if experiment 2 and 3 are underpowered.
It is noted the data was pooled for analysis despite three different exercise protocols. I question the appropriateness of this, due to the different protocols implemented. How can the data be appropriately compared?
Results: The figures are very small and hard to interpret. I understand what this research is trying to achieve, but I am not confident the pooling of results utilising different protocols is appropriate for the development of these models. Perhaps I am misinterpreting the study design somehow? I understand further on in the results section you are grouping based on intensity for the prediction of EE, which does make sense, but the use of different protocols is still not ideal. I think it would be best to present the data for experiment 1 which appeared to have appropriate power, and therefore all participants were completing the same protocols.
Discussion: There is discussion around BMI, but I feel some discussion around the differences in EE considering fat free mass and fat mass is warranted, not just BMI. I also feel we need more information about participants and their level of fitness to see what population more specifically these models could be applied to
Reviewer 2 Report
Comments and Suggestions for Authors
The manuscript explores the predictive accuracy of HR- and HRres-based models for energy expenditure and addresses a relevant question in exercise physiology with practical implications for wearable technologies. The study is well written and supported by an extensive reference base; however, the small and unbalanced sample, reliance on age-predicted HRmax, and the pooling of heterogeneous datasets limit the strength of the conclusions. Furthermore, the models were both developed and validated within the same dataset, raising concerns of potential overfitting. The results presentation is overly detailed in tables and would benefit from clearer graphical summaries. Overall, the study is promising but requires major revisions to strengthen methodological rigour and ensure more balanced conclusions.
Reviewer 3 Report
Comments and Suggestions for Authors
Dear Authors,
Thanks for your submission. Unfortunately, at this stage I cannot endorse this for publication. I found that the manuscript suffers from several fundamental and irremediable flaws in its research design and analytical approach. These issues are sufficiently severe that they undermine the validity of the study's conclusions.
Briefly, the study's dataset is a composite of three separate, distinct experiments, with different participants, each conducted with a different exercise protocol (Bruce, Modified Bruce, and a progressive speed protocol). Pooling these heterogeneous datasets introduces significant uncontrolled variance and confounding variables that are not statistically addressed and it is impossible even with statistical adjustments to compare a group with 4 to a group of 16 when they did different protocols.
Sample size was incorrect calculated, based on correlations, while regressions where applied. This study cannot be replicated.
The conclusions drawn are not supported by the methodological approach. A complete redesign of the study, involving a single, coherent experimental protocol and a separate cohort for model validation, would be necessary to adequately address the research question.

Reviewer 4 Report
Comments and Suggestions for Authors
Lines 68 and 69: What is the justification for having these three groups?
Line 86: The format of Table 1 could be improved. It could be reported if there were differences by gender or between groups.
Line 89: Repeated words were identified; the wording could be improved.
Line 103: VO2 (insert the 2 in subscript).
Line 111: The number 3 should be written in letters.
Line 145: Figure 1 needs to be improved so it can be better viewed.
Comments on the Quality of English LanguageYou can improve the English writing, specifically where indicated in the comments.
